# Genome-wide identification of natural RNA aptamers in prokaryotes and eukaryotes

Sidika Tapsin[1], Miao Sun[2], Yang Shen[2], Huibin Zhang[3], Xin Ni Lim[1], Teodorus Theo Susanto[1], Siwy Ling Yang[1], Gui Sheng Zeng[4], Jasmine Lee[4], Alexander Lezhava[5], Ee Lui Ang[3], Lian Hui Zhang[4], Yue Wang [4], Huimin Zhao [3,6], Niranjan Nagarajan[2] & Yue Wan[1]

RNAs are well-suited to act as cellular sensors that detect and respond to metabolite changes in the environment, due to their ability to fold into complex structures. Here, we introduce a genome-wide strategy called PARCEL that experimentally identifies RNA aptamers in vitro, in a high-throughput manner. By applying PARCEL to a collection of prokaryotic and eukaryotic organisms, we have revealed 58 new RNA aptamers to three key metabolites, greatly expanding the list of natural RNA aptamers. The newly identified RNA aptamers exhibit significant sequence conservation, are highly structured and show an unexpected prevalence in coding regions. We identified a prokaryotic precursor tmRNA that binds vitamin B2 (FMN) to facilitate its maturation, as well as eukaryotic mRNAs that bind and respond to FMN, suggesting FMN as the second RNA-binding ligand to affect eukaryotic expression. PARCEL results show that RNA-based sensing and gene regulation is more widespread than previously appreciated in different organisms.

[1] Stem Cell and Development Biology, Genome Institute of Singapore, Singapore 138672, Singapore. [2] Computational and Systems Biology, Genome Institute of Singapore, Singapore 138672, Singapore. [3] Metabolic Engineering Research Laboratory (MERL), Science and Engineering Institutes, Agency for Science, Technology, and Research (A*STAR), 31 Biopolis Way, Nanos #01-01, Singapore 138669, Singapore. [4] Institute of Molecular and Cell Biology, Proteos, 61 Biopolis Drive, Singapore 138673, Singapore. [5] Translational research group, Genome Institute of Singapore, Singapore 138672, Singapore. [6] Department of Chemical and Biomolecular Engineering, University of Illinois at Urbana-Champaign, Urbana, IL 61801, United States. These authors contributed equally: Sidika Tapsin, Miao Sun, Yang Shen, Huibin Zhang. Correspondence and requests for materials should be addressed to N.N. (email: nagarajann@gis.a-star.edu.sg) or to Y.W. (email: wany@gis.a-star.edu.sg)

Microorganisms are constantly sensing their environment for changes in temperature, pH, metabolites, and nutrients so as to regulate their gene expression programs to best adapt to different signals for growth, survival, and virulence[1]. As such, comprehensive mapping of regulatory networks in microbes under different environmental conditions is crucial to understanding their biology. While many of these regulators have been extensively studied at the protein and DNA levels, RNA's role as direct sensors and responders remains relatively under-explored. One class of cellular RNA sensors, known as riboswitches, can recognize and respond to specific metabolites by altering gene expression[2]. Upon binding to their ligands, riboswitches undergo conformational changes that result in the regulation of gene expression through diverse means, such as changes in transcription, translation and decay[2], informing the host of its environmental conditions. The modularity of riboswitches also allows them to be transplantable to different systems, broadening their use as biological sensors.

While in vitro selection methods, such as SELEX, have been applied with variable success to generate new synthetic RNA aptamers[3], the ability to comprehensively identify natural RNA aptamers from transcriptomes would expand our toolbox for synthetic biology and deepen our understanding of RNA -based gene regulation in vivo. Currently, potential natural RNA aptamers are mostly identified through computational determination of sequence and structural homology to known riboswitches[4]. However, as RNA can adopt different folds for binding to the same ligand and organisms can diverge greatly in sequence content, computational means of searching for riboswitches through sequence homology have limited scope[5], and strategies that allow direct experimental detection are highly desirable. One recent strategy, Term-seq, utilizes high-throughput sequencing to detect differential transcription termination events in bacteria under different conditions[6]. However, complementary strategies for detecting riboswitches that act through other mechanisms (such as translation inhibition), as well as riboswitches that bind to metabolites whose intracellular concentrations are not easily altered, need to be developed.

Here, we report an in vitro method for experimentally identifying RNA aptamers in transcriptomes by detecting ligand-induced RNA structural changes using high-throughput sequencing (Fig. 1a). This method allows us to rapidly screen through transcriptomes to identify natural RNA aptamers toward almost any ligand of choice. Specifically, we extract total RNA from organisms grown under different conditions and probe their structures in the presence or absence of different metabolites by using a double-stranded nuclease, RNase V1, which recognizes and cleaves at base-paired regions in RNAs. The different cleavage sites, in the presence and absence of metabolites, are cloned into cDNA libraries for deep sequencing. Subtle differences between these two libraries point to the few true ligand-specific structure changing RNA elements in the genome and we developed a sensitive and robust computational analysis pipeline to identify these (Methods). This experimental and computational approach, termed Parallel Analysis of RNA Conformations Exposed to Ligand binding (PARCEL), revealed the breadth of RNA-ligand interactions in prokaryotic and eukaryotic transcriptomes, identifying many new natural RNA aptamers in the process.

## Results

**Development of PARCEL.** To establish PARCEL, we systematically tested different structure-probing strategies to determine the approach that best captures ligand-induced structural changes genome-wide, allowing for a simplified, cost-effective workflow without multiple probing assays[7]. Strategies using double-strand or single-strand specific nucleases (RNase V1 and S1 nuclease, respectively), as well as in-line probing which probes nucleotide flexibility[8], were tested for their abilities to detect structural changes using high-throughput sequencing (Fig. 1b, Supplementary Fig. 1, 2, 3). The known thiamine pyrophosphate (TPP) and S-adenosylmethionine (SAM) riboswitches were used as positive controls and other RNA sequences not known to bind TPP or SAM were used as negative controls in this experiment[9,10]. As expected, the known riboswitches showed large structural changes upon ligand binding (Fig. 1b, Supplementary Fig. 1, 2), while the negative controls did not (Supplementary Fig. 3), indicating that the structure changes captured by nuclease digestion followed by sequencing are highly specific. These structural differences can be less pronounced in libraries prepared using in-line probing (Fig. 1b, Supplementary Fig. 2). This is likely due to the noise introduced by the additional 5′ phosphorylation step that is used in in-line probing library preparation. As degraded cellular RNAs are also phosphorylated and cloned into the library, it is difficult to distinguish in-line probed fragments from degradation fragments. Among the nucleases, we observed a higher degree of correlation between two biological replicates of RNase V1 ($R = 0.99$) versus S1 nuclease libraries ($R = 0.66$, Supplementary Fig. 4a), a key feature for differential analysis. The structural changes observed using the nucleases could also be reproduced using low-throughput RNA footprinting and mapped to the secondary structure of the TPP riboswitch (Supplementary Fig. 1b-d). Correspondingly, we selected RNase V1 as the probing strategy of choice in all PARCEL experiments to identify natural RNA aptamers genome-wide.

As *Bacillus subtilis* and *Pseudomonas aeruginosa* are bacteria for which many riboswitches are known, we performed structure probing in the presence and absence of key metabolites known to interact with RNAs in their transcriptomes. To maximize our chances of finding RNA aptamers which may only be expressed under specific conditions, we grew the bacteria in rich or minimal media to exponential or stationary phases (Methods). We then extracted total RNA from the pooled bacteria, performed ribosomal depletion to enrich for mRNAs, and did structure probing, followed by deep sequencing. We had two biological replicates for each experiment and obtained more than 7 million reads per replicate (Supplementary Table 1). RNA aptamers that bind specifically to one ligand should not recognize other unrelated ligands. As such, we developed a novel computational pipeline to identify contiguous positions of structural change (to increase signal-to-noise ratio) that show statistically different numbers of reads that indicate base pairing in one metabolite condition but not in others (Supplementary Fig. 4b, Methods). This approach aggregates signals of variation in each base pair across conditions to define ligand-responsive regions using dynamic programming, and combines this with the computation of a BLAST-like $E$-value to identify RNA aptamers with statistical confidence (Methods).

**PARCEL finds known riboswitches in *B. subtilis* and *P. aeruginosa*.** To evaluate PARCEL, we first determined if we could identify known riboswitches that have been previously reported in the literature. We identified 17 out of 20 known riboswitches that interact with key metabolites (TPP, FMN, and SAM) in *B. subtilis* and *P. aeruginosa*, including 4/5 known TPP riboswitches, 2/2 FMN riboswitches, 9/11 SAM riboswitches in *B. subtilis*, as well as 1/1 TPP and 1/1 FMN riboswitches in *P. aeruginosa*[11], highlighting the high sensitivity of the method (85%; Fig. 1c, Supplementary Fig. 5a-d). Furthermore, pair-wise analysis of control libraries that were generated using the same metabolite did not identify any candidate RNA aptamers, indicating that the approach is highly specific to the presence of metabolites

(Supplementary Fig. 5e). We noted that undetected known riboswitches have low numbers of RNase V1 and S1 nuclease reads along their regulons, indicating that they are either poorly expressed or present in nuclease-inaccessible regions (Supplementary Fig. 5f, g).

Besides detecting structural changes upon ligand binding, we also determined whether PARCEL can detect RNA-ligand

interactions quantitatively. As TPP riboswitches bind most strongly to TPP, followed by thiamine and oxythiamine, we tested the sensitivity of PARCEL in detecting RNA structural changes due to differences in ligand binding affinities[9]. Indeed, we observed the strongest structural change in TPP riboswitches upon binding to TPP, followed by thiamine and then oxythiamine (Fig. 1d, Supplementary Fig. 6a). We also treated *B. subtilis*

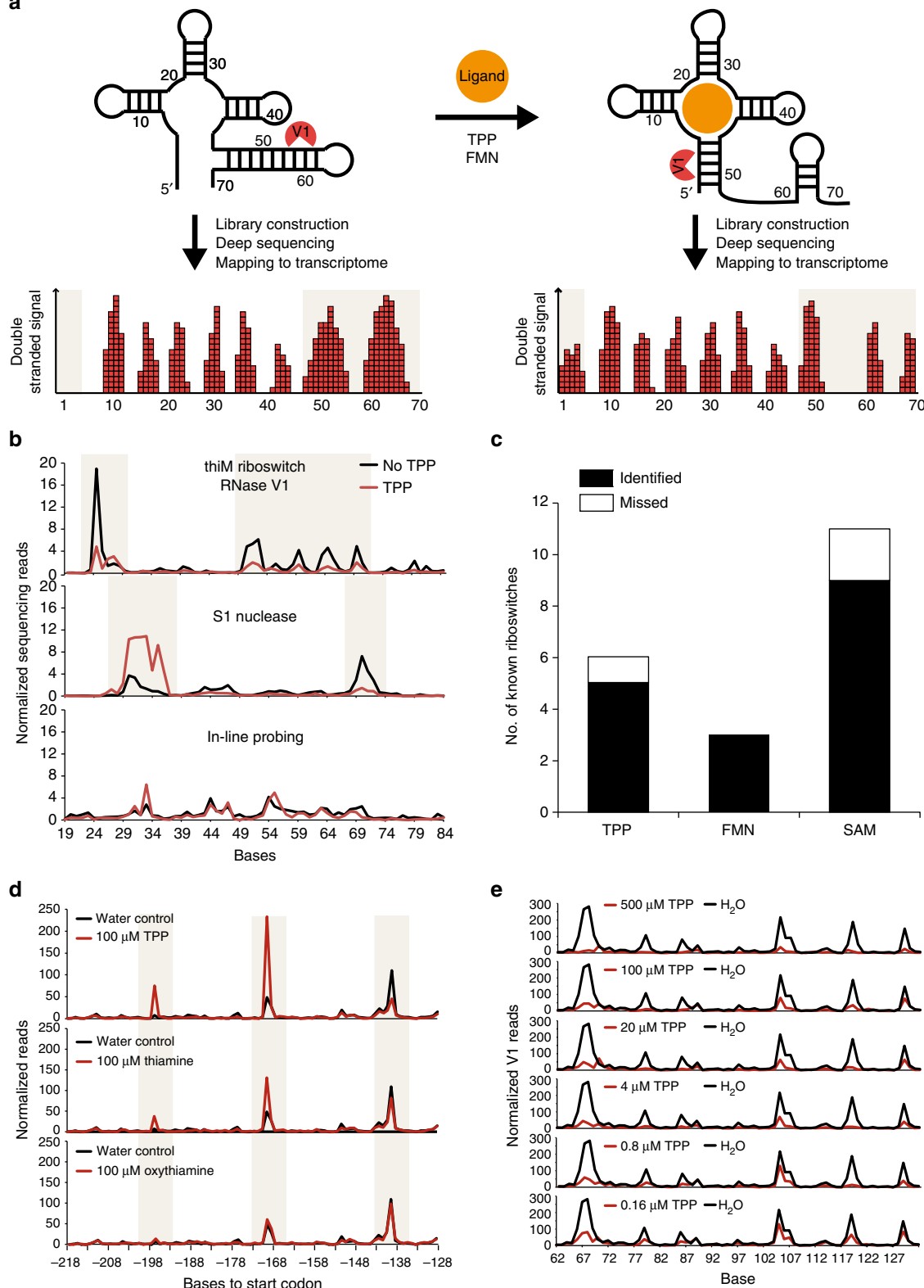

ribosomal RNA-depleted pool with different concentrations of TPP to determine the approximate binding affinity of known TPP riboswitches (Fig. 1e). PARCEL data on the known thiC riboswitch shows a graded change in RNA structure under different ligand concentrations, and an approximate $K_D$ of 110 nM in the most ligand sensitive regions, similar to the previously reported $K_D$ of 100nM[9] (Supplementary Fig. 6b). These data collectively demonstrate that PARCEL is quantitative and can be used to approximate relative binding affinities of RNA-ligand interactions.

**PARCEL identifies new RNA aptamers in *B. subtilis* and *P. aeruginosa*.** Beyond known riboswitches, we also identified new aptamers; 17 TPP and 12 FMN RNA aptamers in the *B. subtilis* and *P. aeruginosa* transcriptomes, as well as 6 SAM RNA aptamers in the *B. subtilis* transcriptome (Fig. 2a, Supplementary Table 2). RNA footprinting validated four out of six ligand-induced structure changes identified by PARCEL (Supplementary Fig. 7a-d). While known riboswitches are mostly located upstream of their operons, we observed that 28% and 47% of the newly identified *B. subtilis* and *P. aeruginosa* RNA aptamers are found in coding regions, respectively (Fig. 2b, Supplementary Fig. 8). Two of our validated RNA aptamers fall in coding regions (Supplementary Fig. 7a, c), indicating that these coding regions exhibit real structural changes in the presence of TPP.

To study the structural properties of these RNA aptamers, we utilized the program Alifoldz[12] to calculate folding energies across their RNA orthologs in different bacterial species (Methods). The newly identified elements exhibit lower folding energies than their dinucleotide shuffled controls, indicating that they are more structured (Fig. 2c). As functionally important RNAs frequently show evolutionary constraints in their sequences, we calculated the nucleotide substitution rate of our new RNA aptamers, as compared to other synonymous regions and UTRs. Similar to known riboswitches, the novel RNA aptamers show a significant reduction in nucleotide substitution rate (Fig. 2d, e), suggesting that they are evolutionarily conserved and likely to be functional.

We observed that one of our prokaryotic metabolite-sensitive regions encodes a small non-coding RNA, specifically a tmRNA (Fig. 3a). The *B. subtilis* tmRNA is transcribed from two promoters to produce both long and short precursor RNAs, which are then processed into the mature tmRNA. To understand the functional role of metabolite sensing in tmRNA, we cloned and in vitro transcribed all three isoforms of tmRNA, then performed structure probing in the presence and absence of FMN in the dark, to avoid FMN-induced photocleavage of RNAs. Interestingly, we observed FMN-induced structure changes in both precursor forms of tmRNA, but not in the mature form (Fig. 3b, c, d, Supplementary Fig. 7d), highlighting that it is the precursor tmRNA that responds to FMN. To determine whether

FMN binding influences the processing of precursor tmRNAs, we grew *B. subtilis* in minimal media with and without the FMN precursor, riboflavin. Addition of riboflavin resulted in a decrease in precursor tmRNA levels and a two-fold increase in mature tmRNA levels (Fig. 3e), supporting the hypothesis that FMN regulates RNA maturation by binding and altering precursor tmRNA structures.

**PARCEL finds new eukaryotic FMN aptamers in *Candida albicans*.** To date, only riboswitches that bind to TPP have been found in eukaryotes, and they regulate splicing and 3′ UTR usage[13,14]. Identifying new eukaryotic riboswitches is important for broadening our understanding of eukaryotic gene regulation. To maximize our chances of finding eukaryotic riboswitches, we screened the fungal pathogen, *C. albicans*, using a pool of metabolites that correspond to highly abundant classes of riboswitches in bacteria, including FMN, SAM, glycine, lysine, and vitamin B12 (Adocbl). PARCEL identified 23 new RNA aptamers that exhibited structural changes in the presence of the metabolite pool (Supplementary Table 3). 87% of the new *C. albicans* RNA aptamers reside in coding regions (Fig. 4a, Supplementary Fig. 8), in contrast to known riboswitches and new prokaryotic aptamers, indicating that they may have different functions from classical riboswitches. We validated seven out of nine PARCEL-identified structural changes by performing in-line probing of these novel RNA aptamers in the presence of the pooled metabolites (Supplementary Fig. 9–11), all of which fall in the coding regions of these genes, confirming that the PARCEL-detected structural changes are real. Similar to their prokaryotic counterparts, the eukaryotic RNA aptamers were found to be significantly more structured compared to dinucleotide shuffled controls, suggesting that structure is likely to be important for their function (Fig. 4b). As many of the new eukaryotic RNA aptamers are located in highly conserved coding regions, we observed an expected increase in conservation of these elements as compared to UTRs, but not a further reduced nucleotide substitution rate compared to other coding sequences (Fig. 4c).

**Eukaryotic RNA aptamers undergo gene expression changes with FMN.** To better understand the cellular roles of the new eukaryotic RNA aptamers, we performed structure probing in the presence of each individual compound in the metabolite pool on two RNA aptamers identified in the coding regions of the genes RPS31 and ATP1. Interestingly, structure probing of these aptamers revealed that they respond specifically to FMN, and not to other metabolites in the solution (Supplementary Fig. 10a, 11a). Detailed structure probing, in the dark, along the length of these two RNAs identified several regions that changed structure in the presence of FMN (Fig. 4d, Supplementary Fig. 10b), suggesting that FMN binding results in structural remodeling of these RNAs. To determine whether changes in the intracellular concentration

**Fig. 1** Measuring RNA-ligand binding by structure probing and deep sequencing. **a** RNA undergoes structure changes upon ligand binding. This structural change is detected by the double-strand specific nuclease, RNase V1, which cuts at different double-stranded places along the RNA in the presence and absence of the ligand. The cleavage sites are then captured and cloned into a cDNA library for deep sequencing. After mapping the reads to the transcriptome, we can identify which bases have undergone changes in structuredness upon ligand binding (highlighted in beige boxes). **b** Deep sequencing reveals structure changes of a known TPP riboswitch, thiM, using RNase V1 (top), S1 nuclease (middle), and in-line probing (bottom). The red and black lines indicate the structure profiles of thiM treated with and without 100 μM TPP, respectively. The beige regions highlight regions of structural changes upon ligand binding. **c** PARCEL identified 85% of known TPP, FMN, and SAM riboswitches in *B. subtilis* and *P. aeruginosa*. The black and the white bars indicate the number of known riboswitches that were captured and missed in our study, respectively. **d** PARCEL sequencing data for the *B. subtilis* TPP riboswitch, thiT, in the presence and absence of 100 μM TPP (top), 100 μM thiamine (middle), and 100 μM oxythiamine (bottom). PARCEL detected strongest structural change in thiT in the presence of TPP, followed by thiamine and then oxythiamine, which corresponds to the binding affinities of TPP riboswitches for these metabolites[9]. **e** The plots show normalized V1 read counts of the thiC TPP riboswitch under increasing concentrations of TPP. PARCEL was performed on the *B. subtilis* transcriptome

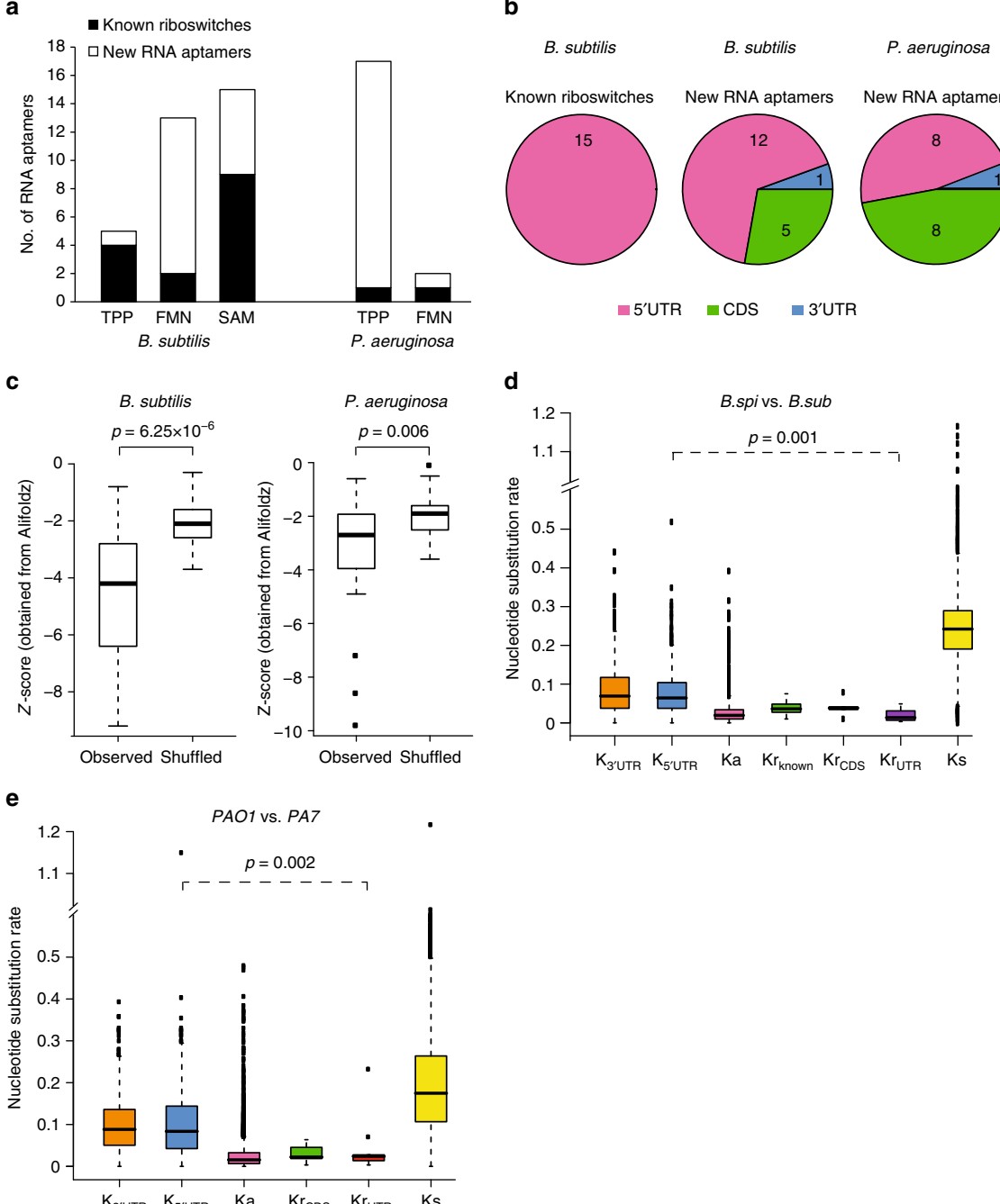

**Fig. 2** PARCEL identifies new RNA aptamers in bacterial species. **a** PARCEL identifies a total of 52 RNA aptamers in *B. subtilis* and *P. aeruginosa*. Black and white bars indicate the numbers of known riboswitches and novel aptamers that are identified in our study, respectively. **b** Distribution of known riboswitches and new RNA aptamers along the 5′ UTR, CDS, and 3′ UTR regions for *B. subtilis* and *P. aeruginosa*, showing that a substantial proportion of RNA aptamers are located in the 3′ UTR and CDS regions. **c** Comparison of score distribution of Alifoldz[12] for RNA aptamers vs. shuffled counterparts. The upper, middle, and lower bounds of the boxplot represent the 75, 50, and 25th percentile of the values, respectively. A negative score indicates a stable, conserved consensus structure. *p*-value was obtained using the non-parametric Kolmogorov–Smirnov test. **d, e** Comparison of the nucleotide substitution rate (number of substitutions per base-pair) for new RNA aptamers in coding region ($Kr_{CDS}$), new RNA aptamers in UTR ($Kr_{UTR}$), 3′ UTR ($K_{3'UTR}$), 5′ UTR ($K_{5'UTR}$), synonymous sites (Ks), and non-synonymous sites (Ka). The upper, middle, and lower bounds of the boxplot represent the 75, 50, and 25th percentile of the values, respectively. To calculate nucleotide substitutions, *B. subtilis 168* was compared to *B. subtilis subsp. spizizenii W23* (**d**), and *P. aeruginosa PAO1* was compared to *P. aeruginosa PA7* (**e**). Note that $Kr_{known}$ denotes the substitution rate of known riboswitches in *B. subtilis* (15 in total) as annotated in the RegPrecise database[11]. $Kr_{known}$ was not calculated in *P. aeruginosa* as there are too few known TPP and FMN riboswitches. *p*-values were calculated using the non-parametric Kolmogorov–Smirnov test

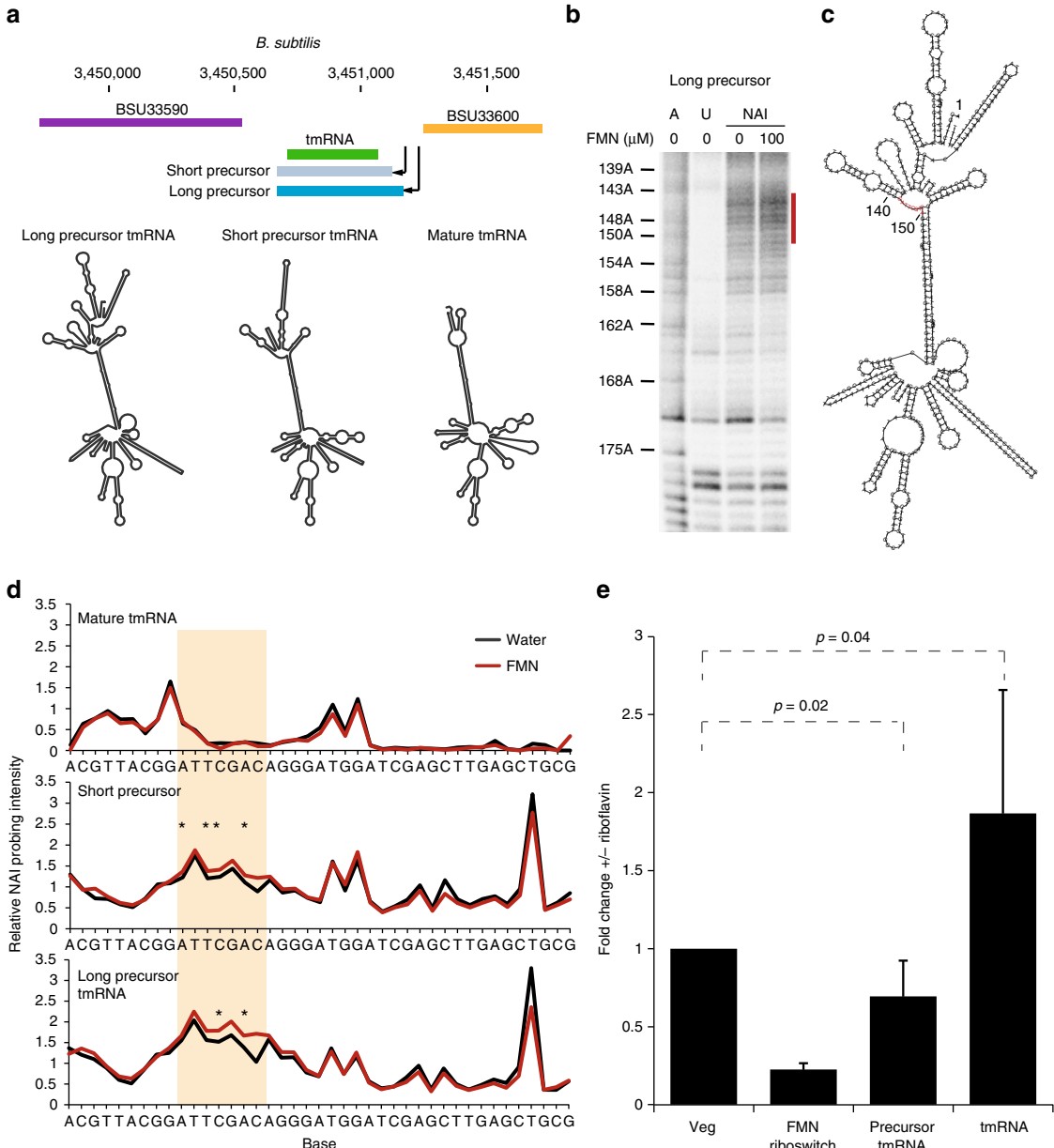

**Fig. 3** Precursor tmRNA can act as an RNA sensor for FMN. **a** Schematic of the *B. subtilis* tmRNA genomic locus (top) and predicted secondary structures of the long precursor tmRNA, short precursor tmRNA, and mature tmRNA, using the RNAfold program[20] (bottom). **b** RNA footprinting analysis of the long precursor tmRNA, using a SHAPE-like chemical (NAI), in the presence (lane 4) and absence (lane 3) of 100 μM FMN. Also shown are A ladder (lane 1) and unmodified RNA (lane 2). The red bar indicates bases that become more single-stranded in the presence of FMN. **c** Predicted secondary structure of the tmRNA long precursor using the RNAfold program[20]. The red bases correspond to the positions marked by the red bar in **b**. **d** Average footprinting analysis (*n* = 3, SAFA) of mature (top), short precursor (middle), and long precursor tmRNA (bottom), in the presence (red) and absence (black) of 100 μM FMN, in the dark. The beige box indicate the region of increased flexibility in the precursor tmRNAs in the presence of FMN. The stars indicate bases that show statistically significant changes with FMN ($p \leq 0.05$, Student *t*-test). **e** qPCR analysis of the mRNA expression level of precursor tmRNA and mature tmRNA, across six biological replicates, after addition of 100 μM of riboflavin to the growth media of *B. subtilis*. Fold changes are normalized to the negative control Veg gene. The known *B. subtilis* FMN riboswitch is used as the positive control. *p*-values were calculated by Student's *t*-test, the error bars indicate standard deviation of the replicates

of FMN could alter gene expression changes of RPS31 or ATP1 in vivo, we integrated FLAG-tagged *C. albicans* RPS31 or ATP1 in a *S. cerevisae* FMN1 synthase deletion mutant (*fmn1Δ*) as *S. cerevisae* is known to take up exogenous FMN, unlike *C. albicans*[15]. Both transcript and protein levels of FLAG-tagged *C. albicans* RPS31 and ATP1 were measured, following growth of the integrated strains under varying FMN concentrations. We

found that while transcript levels of FLAG-tagged *C. albicans* RPS31 and ATP1 in *fmn1Δ* did not change with increasing FMN concentrations (Supplementary Fig. 10c, 11b), RPS31 and ATP1 protein levels were decreased and increased, respectively (Fig. 4e, Supplementary Fig. 10d-f, 11c), suggesting that FMN sensing could have a regulatory effect on gene expression in vivo, and that

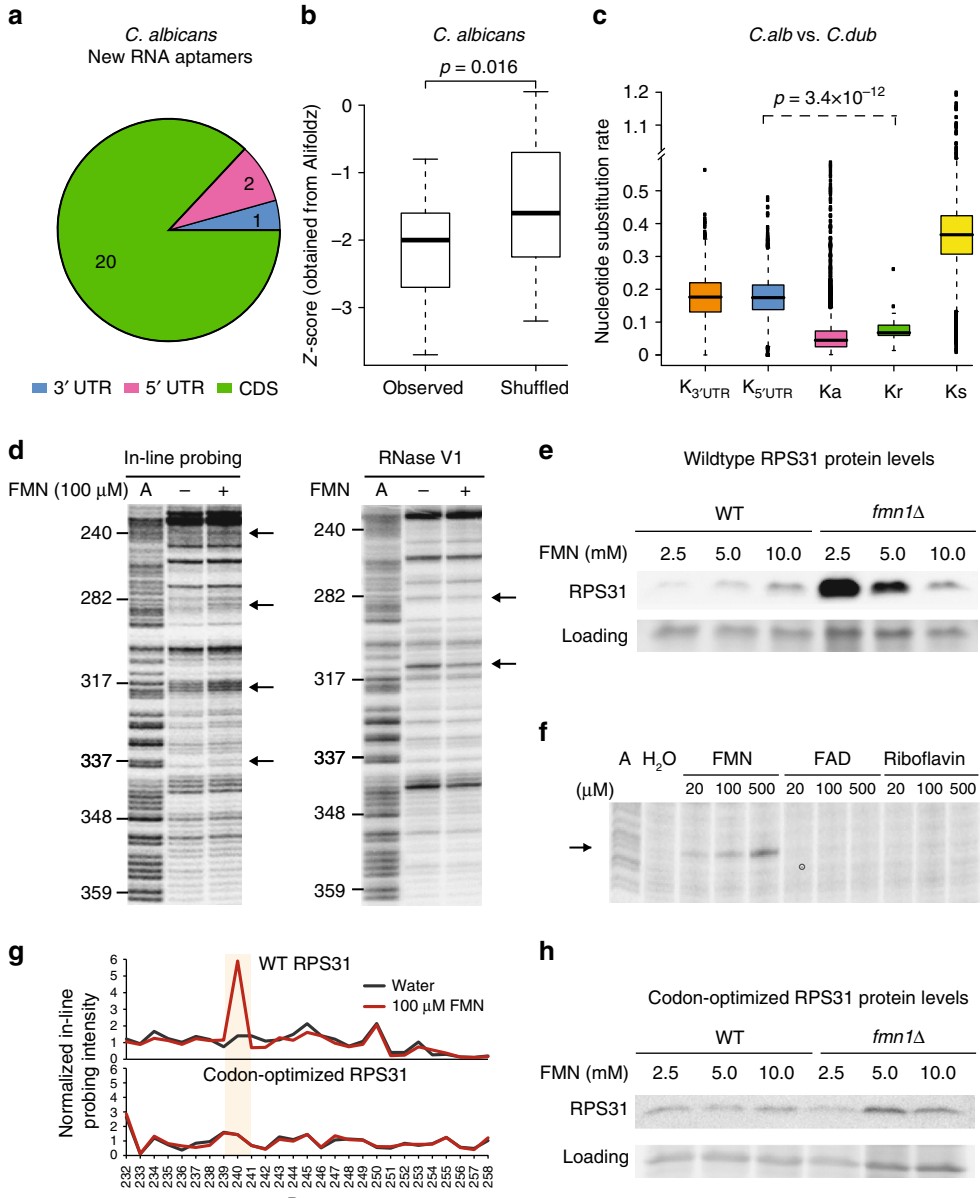

**Fig. 4** PARCEL identifies new RNA aptamers in *Candida albicans*. **a** Pie chart of the number of *C. albicans* RNA aptamers that are located in 5′ UTR, CDS, and 3′ UTR. The majority of *C. albicans* RNA aptamers are found in CDSs. **b** Comparison of the distribution of Alifoldz scores for RNA aptamers vs. shuffled counterpart. The upper, middle, and lower bounds of the boxplot represent the 75, 50, and 25th percentile of the values, respectively. A negative score indicates a stable, conserved consensus structure. *P*-value was obtained using the non-parametric Kolmogorov–Smirnov test. **c** Nucleotide substitution rates, calculated as the number of substitutions per base-pair, for RNA aptamers (Kr), 3′ UTR ($K_{3'UTR}$), 5′ UTR($K_{5'UTR}$), synonymous sites (Ks), and non-synonymous sites (Ka). The upper, middle, and lower bounds of the boxplot represent the 75, 50, and 25th percentile of the values, respectively. *C. albicans SC5314* was compared to *Candida dubliniensis* for the calculation. *p*-value was obtained using the non-parametric Kolmogorov–Smirnov test. **d** Gel analysis of RPS31 mRNA using in-line probing (left) and RNase V1 (right) in the presence (lane 3) and absence (lane 2) of 100 μM FMN. The A ladder (A, lane 1) is also shown. The black arrows indicate positions along the RNA that changed in the presence of FMN. **e** A representative Western blot showing RPS31:: FLAG (top) and loading (bottom) protein levels in RPS31::FLAG knock-in strains with WT (left) and *fmn1Δ* (right) backgrounds cultured at different FMN concentrations (mM). Using *t*-test (*n* = 8), significant *p*-values of 0.009 and 0.01 (for 2.5 and 5.0 mM against 10.0 mM, respectively) were determined for *fmn1Δ*, but not WT (*p*-values of 0.2 and 0.5). **f** Gel analysis of RPS31 mRNA using in-line probing in the presence of 20, 100, or 500 μM of FMN, FAD or riboflavin. In-line probing of RNA in the absence of metabolite ($H_2O$, lane 2) and A ladder (A, lane 1) are also shown. **g** SAFA analysis of WT RPS31 (top) and codon-optimized RPS31 (bottom) in the presence (red line) and absence (black line) of 100 μM FMN. The beige box indicates the region of structural change in WT RPS31 when it interacts with FMN. This structural change is absent in the codon-optimized RPS31. **h** A representative Western blot showing codon-optimized RPS31::FLAG (top) and loading (bottom) protein levels in codon-optimized RPS31::FLAG knock-in strains with WT (left) and *fmn1Δ* (right) backgrounds cultured at different FMN concentrations (mM). Using *t*-test (*n* = 3), calculated *p*-values for 2.5 and 5.0 mM were insignificant for both *fmn1Δ* (both 0.7) and WT (0.9 and 0.09)

these genes could represent the first known eukaryotic FMN riboswitches.

To further understand ligand binding specificities and affinities of this putative FMN riboswitch, we performed detailed structural studies on the RPS31 transcript. We observed that RPS31 RNA binds specifically to FMN and does not respond to structurally similar analogs, such as riboflavin and FAD (Fig. 4f, Supplementary Fig. 11d). Integrating double-stranded (RNase V1), single-stranded (S1 nuclease) structure probing and in-line probing information along the length of the RPS31 transcript into the RNAfold prediction software showed that RPS31 RNA consists of seven stems around a central loop, and that FMN binding results in extensive structural rearrangements (Supplementary Fig. 12a). The FMN-bound RPS31 aptamer consists of six stems around the central FMN bound loop, and appears to resemble the prokaryotic FMN riboswitch structure[16]. This attests to the structural plasticity of RNA molecules, whereby different sequences can be utilized to form similar structures for cellular function.

To further test whether the FMN-induced change in RPS31 protein levels is mediated post-transcriptionally, we designed a codon-optimized version of *C. albicans* RPS31 (by changing nucleotides of synonymous bases) to disrupt RPS31 RNA structure without altering its protein sequence. As expected, codon-optimized RPS31 mRNA is structurally different from wildtype RPS31 mRNA, and does not show structure changes in the presence and absence of FMN (Fig. 4g). Codon-optimized RPS31 maintained similar mRNA and protein levels at varying concentrations of FMN (Fig. 4h, Supplementary Fig.12b-d), supporting the hypothesis that the binding of FMN to native *C. albicans* RPS31 RNA results in post-transcriptional regulation of RPS31 protein levels.

## Discussion

In summary, we have developed a new strategy named PARCEL that experimentally identifies RNA aptamers transcriptome-wide by detecting ligand-induced structure changes. As PARCEL allows us to rapidly screen through transcriptomes to identify RNA aptamers, we identified a total of 58 novel candidate RNA aptamers in two prokaryotic and one eukaryotic species, including a second class of putative eukaryotic riboswitches. Unlike known riboswitches described in the literature, the newly identified aptamers reside in both UTRs and coding sequences, and are not necessarily linked to the biosynthetic pathways of their respective ligands. Further characterization of three new RNA aptamers showed that they could be riboswitches as FMN-sensing induces RNA structural changes and regulates transcript levels of tmRNA in *B. subtilis*, and protein levels of RPS31 and ATP1 in *C. albicans*. As PARCEL can be readily applied to any transcriptome and ligand, we believe that further application of PARCEL to diverse organisms will result in the identification of many novel natural RNA aptamers in the near future, providing new building blocks for biological sensing and deepening our understanding of RNA-based gene regulation in vivo.

## Methods

**Bacterial and yeast cultures**. *P. aeruginosa* PAO1 and *B. subtilis* 168, were grown in LB or minimal media to log ($OD_{600} = 0.6–0.8$) or stationary phases ($OD_{600} > 2$). Total RNA from *P. aeruginosa* was extracted using Trizol reagent (Thermo Fisher Scientific). Total RNA from *B. subtilis* was extracted by first incubating *B. subtilis* in 4 mg per mL of lysozyme for 15 min before using Trizol LS reagent. Ribosomal depleted RNA, Ribo(−) RNA, was obtained by using Ribo-Zero rRNA Removal Kit (Epicenter) according to manufacturer's instructions. *S. cerevisiae* S288C was grown in YPD to exponential phase ($OD_{600} = 0.6–0.8$). *C. albicans* strain SC5314 was grown in YPD or GMM (yeast nitrogen base without amino acids and with 2% glucose) to exponential ($OD_{600} = 0.6–0.8$) or stationary phases ($OD_{600} > 2$). Total RNA from *S. cerevisiae* or *C. albicans* was extracted using a slightly modified protocol that uses hot acid phenol[17]. Poly(A)+ RNA was obtained by using the

Poly(A) Purist MAG kit (Thermo Fisher Scientific) according to manufacturer's instructions. Poly(A)+ or Ribo(−) RNA were then structure probed in the presence and absence of metabolites.

*fmn1Δ* mutant was created by replacement of FMN1 in BY4741 strain with KanMX using homologous recombination[18]. RPS31 from *C. albicans*, with a C-terminal FLAG-tag (GATTACAAGGACGACGATGACAAG), was integrated together with URA3 at the *ura3Δ* site to generate the RPS31::FLAG knock-in strains. ATP1 from *C. albicans*, with a N-terminal FLAG-tag, was integrated together with URA3 at the *ura3Δ* site to generate the ATP1::FLAG knock-in strains.

**RNA structure probing**. Briefly, 250 ng of Poly(A)+ or Ribo(−) RNA was heated to 90 °C for 2 min and cooled on ice for 2 min before adding 10X RNA structure buffer (500 mM Tris pH 7.4, 1.5 M NaCl, 100 mM $MgCl_2$) and metabolites to the RNA. The RNA pool was slowly brought to 37 °C for 30 min and structure probed using RNase V1 (1:2000 dilution, AM2275 Life Technologies) or S1 nuclease (1:500 dilution, Fermentas) at 37 °C for 15 min. The nuclease reactions were inactivated using phenol chloroform extraction and ethanol precipitated. In-line probing reactions were performed in 50 mM Tris-HCl (pH 8.3), 20 mM $MgCl_2$, and 100 mM KCl at 25 °C for 40 hours[8]. The in-line probed RNA was phosphorylated using T4 polynucleotide kinase (PNK) in T4 PNK buffer and 1 mM ATP to capture the cleavage sites.

**Library preparation**. Structure probed RNA was fragmented at 95 °C for 3.5 min in alkaline hydrolysis buffer (Ambion). As fragmentation results in 5′OH, and is hence ligation incompatible, it does not interfere with the downstream library preparation process. Fragmented RNA was then purified using RiboMinus concentration module (Life Technologies), using the modified protocol for RNAs that are <200 bases. The RNA was eluted in 12 µl of nuclease free water and concentrated to 2 µl using a vacuum centrifuge. The RNA was then ligated to 5′ adapter from NEBNext Multiplex Small RNA Library Prep Set for Illumina using T4 RNA ligase1 (T4 RNA ligase buffer, 1 mM ATP, 10% PEG, 10% DMSO) at 16 °C overnight. The 5′ adapter ligated RNAs were then purified through a 6% TBE urea PAGE gel and size selected for 50–200 bases. The RNA was then ligated to 3′ adapter, reverse transcribed, and PCR amplified using the NEBNext Multiplex Small RNA Library Prep Set (New England Biolabs) for Illumina using manufacturer's instructions. Eighteen cycles of PCR amplification were typically performed for each library.

**RNA footprinting analysis**. Cleavage and modification sites along structure probed RNA were identified using primer extension. Briefly, a primer located ~30–50 bases downstream of the structure probed region was labeled with γP32 ATP using T4 PNK kinase. The labeled primer was then purified using a 15% TBE urea PAGE gel. The labeled primer was incubated with the RNA at 65 °C for 5 min, followed by 35 °C for 5 min, and then cooled at 4 °C. To detect the structure probed sites, we add 3 µl of enzyme mix (4:1:1 of first-strand buffer: DTT: NTP) to the reaction, incubated at 52 °C for 1 min, and Superscript III was added to the reaction at 52 °C for 10 min. To generate a sequencing ladder for the RNA, we added 1 µl of ddNTP (5 mM) to the reaction after the enzyme mix, and before adding Superscript III. 4 M sodium hydroxide was added to the reaction to denature the RNA before the samples were loaded onto a 7 M TBE-Urea PAGE sequencing gel. Gel images were quantified using the software Semi-automated footprinting analysis (SAFA)[19].

**RNA structure models**. RNA secondary structure predictions were generated using RNA footprinting data with RNase V1, S1 nuclease, and NAI as constraints, using the program RNAfold[20] with default parameters.

**qPCR and Western blotting for wildtype and codon-optimized RPS31**. The RPS31::FLAG and ATP1::FLAG strains were inoculated from single colonies into 2 mL SC-ura media and grown overnight at 30 °C, with shaking. Strains with the *fmn1Δ* mutation were supplemented with 10 mM FMN and 200 µg per mL G418 in the cultures. The overnight cultures (1:100 dilution of $OD_{600}$ 2.0) were used to seed 50 mL YPD cultures supplemented with 2.5, 5.0 or 10.0 mM FMN. Cells were harvested after 4–6 h of growth at 30 °C, with shaking (when $OD_{600}$ reaches 0.4). The cultures were split for RNA extraction and Western blotting, pelleted, and washed once with PBS. The resultant cell pellets were frozen and stored at −80 °C.

**RNA extraction and qPCR**. RNA was extracted from frozen yeast pellets using the hot acidic phenol method and treated with TURBO™ DNase (ThermoFisher Scientific)[17]. We made cDNA using the Transcriptor First Strand cDNA Synthesis Kit (Roche) and qPCR was performed using SYBR Green Master Mix (Roche) on a Light Cycler 96 instrument (Roche). Primers used are listed as below. The RPS31 and ATP1 primers are specific for the knock-in *C. albicans* RPS31 and ATP1, and do not amplify the endogenous *S. cerevisiae* RPS31 and ATP1. Normalized fold changes were calculated by normalizing against actin (ACT1) and the respective strain cultured at 10 mM FMN.

|  | **Forward** | **Reverse** |
|---|---|---|
| RPS31 set 1 | (HZ_pri072) TCCACC AGACCAAC AAAGATTG | (HZ_pri073) ACCAAGTG CAAGGTGGATTC |
| RPS31 set 2 | (HZ_pri023) GAATCCA CCTTGCACTTGGTC | (HZ_pri024) GCCAACT TGTGTTTTCTGTGC |
| RPS31 set 3 | (HZ_pri015) GCACAGAA AACACAAGTTGGC | (HZ_pri016) CCATGAAA ATACCGGCACCAC |
| ACT1 | (HZ_pri051) ATGGATTC TGAGGTTGCTGC | (HZ_pri052) TGGTCT ACCGACGATAGATGG |
| RPS31_codon-optimized set 1 | (HZ_pri118) TGGAGGTCG AGTCATCAGATAC | (HZ_pri119) CGTCTTA TTTTCGCAGGGAAGC |
| RPS31_codon-optimized set 2 | (HZ_pri122) TTTCGCA GGGAAGCAGTTAG | (HZ_pri123) TTTTTCCC CCACCCCTTAACC |
| RPS31_codon-optimized set 3 | (HZ_pri120) TAGAGAG GTTGAGGCGTGAATG | (HZ_pri121) GGCACTT ACCGCAATATTGACG |
| ATP1 set 1 | (HZ_pri064) TTACGTA CTGCTGCTCGTACAG | (HZ_pri065) GGCAGAG GCAAATCTTTGAGC |
| ATP1 set 2 | (HZ_pri058) AAGTCG GGGTTGTGTTGTTC | (HZ_pri059) TTCTGGA CCAATTGGAACGG |
| ATP1 set 3 | (HZ_pri062) TCGCTGG TGTTAACGGTTTC | (HZ_pri063) CACCCTT GGTCTTAATAGCATCC |

**Western blotting**. The frozen cell pellet was resuspended in 80 μL of lysis buffer (50 mM Tris pH 7.4, 4% SDS) with proteinase inhibitor (one tablet of complete™, Mini, EDTA-free [Roche] in 2.5 mL of lysis buffer). An equal volume of glass beads (425–600 μm, Sigma-Aldrich) was added and cells were lysed in a Mini-Beadbeater-96 (Biospec Products) for two cycles of 15 s with a two minute interval. Cell lysates were centrifuged and supernatants were run on 4–20% Mini-Protean TGX Stain-Free protein gels (Bio-Rad). To determine the relative levels of total protein (loading), the gels were first imaged on a ChemiDoc MP System (Bio-Rad) using the stain-free technology[21]. Following wet transfer and blocking with 5% milk in PBST, RPS31::FLAG and ATP1::FLAG were detected using mouse anti-FLAG M2 primary antibody (1:4000 in PBS at 4 °C overnight, Sigma-Aldrich, F1804) and sheep anti-mouse IgG, HRP-linked secondary antibodies (1:20000 in 1% milk at 25 °C for an hour, GE Healthcare, NA931). All images were taken using the ChemiDoc MP System (Bio-Rad) and analyzed with ImageJ[22]. Corresponding uncropped images of blots (in main figures) can be found in supplementary figures.

**Read mapping**. Short reads from PARCEL libraries (Illumina HiSeq, 50 bp, single-end) in *P. aeruginosa* PAO1, *B. subtilis* 168, and *E. coli* K12 were aligned to their corresponding reference genomes downloaded from NCBI using the short-read aligner bowtie2 (parameters: −k 1–local)[23]. In the case of *S. cerevisiae* S288C and *C. albicans* SC5314, we extracted the UTR annotation from Bruno et al.[24,25] and integrated them into their corresponding transcriptomes before alignment by bowtie2. In both cases, only uniquely mapped reads were used for subsequent analysis.

**Identification of RNA aptamers**. For each position along the genome or transcriptome, we counted the number of reads whose first mapped base was one base downstream of the inspected position. Higher counts suggest greater accessibility to V1 nuclease, and are more likely to be associated with a double-stranded conformation. In all expressed transcripts, positions with zero count could either be associated with a single-stranded conformation, or come from a heavily folded region that is inaccessible to V1 nuclease.

Since RNA structural changes should typically span across multiple bases, we looked for regions that exhibit differential V1 counts to increase the sensitivity/specificity of detecting RNA aptamers. We first evaluated the significance of differential V1 counts at each nucleotide position using the edgeR package[26], where we compared samples treated by one specific metabolite (e.g., TPP) to samples from all other conditions. We focused on positions that were generally accessible to the V1 nuclease by applying a minimum abundance threshold (average counts per sample per position, $a > 1$) and then computed a score $s_i$ for each passed position $i$ based on edgeR-generated p-values (pval$_i$): $s_i = \ln(0.1) - \ln(\text{pval}_i)$ (in effect, giving negative scores for p-values > 0.1).

The higher the score, $s_i$, the more likely that differential V1 cutting was observed at that specific position. Accordingly, positions that failed to pass the abundance threshold were assigned a penalty score of −10. We then looked for segments of contiguous positions (e.g., a segment from position m to n) with the highest aggregate score $S = \sum s_i$, by applying the Kadane algorithm (maximal subarray problem). We then determined the significance of these high-scoring segments based on Karlin–Altschul statistics, similar to the approach used in BLAST[27].

As described by Karlin and Altschul[27], the expected value (E-value) of high-scoring segments with an aggregate score of at least $S$ is given by the formula:

$$E_v = Ke^{-\lambda S}. \tag{1}$$

Therefore, we examined the extreme value distribution of the aggregate score $S$ to estimate the two parameters required i.e., $K$ and $\lambda$. Specifically, $\lambda$ can be calculated from the formula: $\sum p_i e^{\lambda s_i} = 1$, where $p_i$ is the corresponding probability of the scores$_i$. As $s_i = \ln(0.1) - \ln(\text{pval}_i)$ and pval$_i$ approximately follows a uniform distribution (U(0,1)) due to the assumption that the majority of nucleotide positions do not undergo any structural changes, the equation $\sum p_i e^{\lambda s_i} = 1$ can be translated to:

$$\int_{\text{pval}_i=0}^{1} e^{\lambda(\ln(0.1)-\ln(\text{pval}_i))} = 1.$$

This can be solved to $\frac{0.1^\lambda}{1-\lambda} = 1$, and $\lambda = 0.862871$. The parameter $K$ is bounded between $K^- = C^*\left(\frac{\lambda\delta}{e^{\lambda\delta}-1}\right)$ and $K^+ = C^*\left(\frac{\lambda\delta}{1-e^{-\lambda\delta}}\right)$. Since $\delta$ is the smallest span of $s_i$, $K$ is bounded between $\lim_{\delta\to 0} K^-$ and $\lim_{\delta\to 0} K^+$ and $C^*$ is defined by the formula:

$$C^* = \frac{e^{-2\sum_{k=1}^{\infty}\frac{1}{k}\left(E\left(e^{\lambda S_k}; S_k<0\right)+\text{Prob}(S_k\geq 0)\right)}}{\lambda E(S_1 e^{\lambda S_1})}.$$

Here, $S_k$ is the random variable representing the sum of $k$ independently chosen $s_i$ i.e., $S_k = \sum s_i = k(\ln(0.1)) + \sum(-\ln(\text{pval}_i))$. Since pval$_i \sim$ U(0,1) $\sum(-\ln(\text{pval}_i))$ approximately follows the gamma distribution i.e., $\Gamma(k=k, \delta=1)$. Let $X_k = \sum_{i=1}^{k}(-\ln(\text{pval}_i))$, it can then be derived that:

$$E\left(e^{\lambda S_k}; S_k<0\right) = 0.1^{\lambda k}\int_{X_k=0}^{-k\ln 0.1}\frac{X_k^{k-1}e^{(\lambda-1)X_k}}{(k-1)!},$$

$$\text{Prob}(S_k \geq 0) = 1 - \frac{\int_{t=0}^{-k\ln 0.1}t^{k-1}e^{-t}}{(k-1)!},$$

and

$$\lambda E\left(S_1 e^{\lambda S_1}\right) = \lambda 0.1^\lambda \frac{1-(\lambda-1)\ln 0.1}{(\lambda-1)^2}.$$

Taken together, $C^*$ can be solved to take the value of 0.0809635, and the upper and lower bounds of $K$, $K^-$, and $K^+$, both equal $C^*$, i.e., $K = 0.0809635$. We then calculated $E_v$ for high-scoring segments by applying equation (1). Segments that pass the $E_v$ threshold of 10 were considered as candidate RNA aptamers that undergo metabolite-responsive conformational changes. Under all conditions, we report candidate regions that have positions with absolute fold-change $f > 2$, relative abundance greater than median + standard deviation for the transcript (abundance-filter; to avoid segments with lower accessibility) and low bonferroni-corrected p-value (<10; to avoid segments with no strongly changing position).

**Distribution of RNA aptamers across operons and transcripts**. We evaluated the distribution of RNA aptamers across operons in bacteria, and along transcripts in fungi (including a 500 bp window on either side when UTRs were not specified). We plotted the histogram of all RNA aptamer positions, with operons in bacteria and coding regions in fungi being scaled to 1 kbp. There are cases where the same position can be considered as belonging to multiple classes and in such cases, we preferentially assigned positions to the 5′ UTR, then to the operon or CDS, and lastly to the 3′ UTR.

**Sequence conservation of RNA aptamers**. We estimated the sequence conservation of identified RNA aptamers by measuring nucleotide substitution rate of these regions to their blastn-identified orthologous sequences. If the identified aptamer regions were shorter than 200 bases in length, we extended them on both sides to a maximum of 200 bases. As highly divergent and highly similar sequences would result in an unreliable estimate of nucleotide substitution rate[28], we chose fairly divergent, and yet not too divergent species (median Ks ranges from 0.1 to 0.4) for this analysis. The orthologous riboswitches, 3′ UTR, 5′ UTR, and protein coding regions were identified using blastn, for non-coding, or genblastG[29], for coding sequence, in other species, respectively. To identify orthologous noncoding sequences in other organisms with high sensitivity, we changed the default blastn parameters as follows: "-e 1e-5 -word_size 7 –gapopen 2 –gapextend 1"[30]. We aligned the noncoding sequences using MUSCLE[31], and the coding sequences using MACSE[32], to construct the multiple sequence alignment. The nucleotide substitution rate of riboswitches, 3′ UTR and 5′ UTR were calculated using Kimura's 2-parameter method[33]. The synonymous and non-synonymous substitution rates was calculated using Kakscalculator[34] with the LPB method.

**Calculating the degree of pairedness for RNA aptamers and controls**. We searched for orthologous sequences of RNA aptamers identified in *B. subtilis*, *P. aeruginosa*, and *C. albicans* across the *Bacillus*, *Pseudomonas*, and *Candida* genus using blastn (with parameters: "-e 1e-5 -word_size 7 –gapopen 2 –gapextend 1"). The species that were used in each genus are: *B. subtilis* XF-1, *B. subtilis* BSn5, *B. malacitensis* CR-95, *B. natto* BEST195, *B. licheniformis* DSM13, *B. subtilis* subsp.

*spizizenii* W23, and *B. cereus* ATCC-14579 for *Bacillus*; *P. aeruginosa* PAO1, *P. aeruginosa* PA7, *P. mendocina* 1267_PMEN, *P. knackmussii* B13, *P. oryzihabitans* USDA-ARS-USMARC-56511, *P. pseudoalcaligenes* KF707, *P. stutzeri* A1501, *P. stutzeri* A1501, *P. mendocina* ymp, *P. entomophila* L48, *P. putida* F1, *P. fluorescens* SBW25, and *P. syringae* pv. *tomato str*. DC3000 for *Pseudomonas*; *C. albicans* SC5314, *C. albicans* WO-1, and *C. dubliniensis* for *Candida*. We then built multiple species alignments for each RNA aptamer region using MUSCLE[31]. We used the program Alifoldz[12] to calculate the energy and structural stability of the consensus structure. For each RNA aptamer alignment, a shuffled alignment was generated as a control using the "shuffle.pl" script from the Alifoldz package.

**Data availability**. All relevant data are available from the authors upon request. Data has been deposited under GEO accession number GSE106133.

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

## Acknowledgements

We thank members of the Wan lab, Nagarajan lab, S. Chen, W.F. Burkholder, A. Sim, H. H. Ng, and B. Lim for discussions. *B. subtilis 168* was obtained from the Bacillus Genetic Stock Center. N. Nagarajan is supported by funding from A*STAR. Y. Wan is supported by funding from A*STAR, Society in Science - Branco Weiss Fellowship, and EMBO Young Investigatorship.

## Author contributions

Y.W. conceived the project, developed the protocol, and designed the experiments. N.N. and M.S. designed the computational pipeline. Y.W., S.T., X.N.L., T.T.S., G.S.Z., J.L., Y.W., L.H.Z., E.L.A., H.Z. and H.Z. planned and performed all the experiments. N.N., S.L.Y., and M.S. planned and conducted the data analysis. A.L. helped with the sequencing. Y.W. organized and wrote the paper with contributions from H.Z., S.L.Y., N.N. and all other authors.

## Additional information

**Competing interests:** The authors declare no competing interests.

