## [Peer Review File(PDF 137 kb) · Nature Communications]

Reviewers' Comments:

Reviewer #4:

Remarks to the Author:

Reviewer #4:

Remarks to the Author:

Reviewer 4:

The authors have addressed some of my concerns but here are ones that remain.

1.) Figure 6. I don't see that these data are substantially different in vivo and in vitro. Are there enough replicates to know that this is robust?

Beyond showing that RPS31 changes structure in vitro- we also performed RNA structure probing in vivo to determine structural changes in the presence of FMN. However in vivo targeted RNA structure probing is technically very difficult because RPS31 transcript levels is too low to be detected on a radioactive gel, without amplification. As such, we only managed to obtain one replicate after many attempts. We have removed this data from the manuscript.

REVIEWER 4: It is appropriate that these data have been removed.

2.) Fig 9 is not an appropriate equation for the data. The authors need a curve that doesn't go through the origin but allows a y-intercept of any value, e.g. ~ 1 . Here the 1/2 value will be >100 μM FMN, not 50 μM .

We thank the reviewer for pointing this out and agree that we are ignoring the background signal in the extrapolated curve. Our recalculations show a K_d of $\sim 137\mu\text{M}$ for RPS31-FMN interactions and we have updated this value in the manuscript (Extended data 11c).

REVIEWER 4: This is a step in the right direction and the value looks to be appropriate from graphical analysis of Extended Data 11c. However, the authors need to show data and a smooth fit. What they currently show in Extended Data 11c seems to be a connect-the-dots plot of the data, given it is non-monotonic, with no fit. They say "Our recalculations show a K_d of $\sim 137 \mu\text{M}$..." What do they mean by "recalculation"? Is there any calculation here or just a best guess at where $\frac{1}{2}$ max is? Please replace this with data as points and a smooth fit to a proper K_d theoretical curve using a non-linear fitting program.

4.) This was the main concern and is not addressed. The biology just doesn't calculate.

We thank the reviewer for his comments. We have three examples of gene regulation by FMN (one at the transcriptional level for *B. subtilis* tmRNA, and two at the protein level for *C. albicans* ATP1 and RPS31).

REVIEWER 4: What do the authors mean when they write above that "We have three examples"? Who are the "We" in this sentence? The field or the authors lab? I believe they mean the field and if so the writing here is potentially misleading.

While identifying riboswitches is one of the aims for designing PARCEL, we note that PARCEL is a platform to find naturally occurring RNA aptamers in virtually any transcriptome, towards any metabolite.

REVIEWER 4: The implication in the writing here via "naturally occurring" is that the aptamer is an in vivo aptamer, but the data as is do not support this. To be publishable, this statement should be revised to not imply that these are naturally occurring without direct demonstration in vivo as such.

Some of the new aptamers may be riboswitches with important gene regulatory functions, while others can be used in metabolite engineering to create higher affinity RNA aptamers for intracellular molecular sensing. As such, we believe that detailed mechanistic studies are beyond

the scope of this paper as they will require significant and extended investigations to understand the role of RNA sensors in global cell biology.

Reviewer #5:

Remarks to the Author:

I reviewed this manuscript for Nature Biotechnology, and in addressing my comments during that process I think this paper is acceptable for publication in Nature Communications.

We thank the reviewers for their positive comments, which have improved the manuscript. Here are our point by point response.

Reviewer #4:

Remarks to the Author:

Reviewer 4:

The authors have addressed some of my concerns but here are ones that remain.

1.) Figure 6. I don't see that these data are substantially different in vivo and in vitro. Are there enough replicates to know that this is robust?

Beyond showing that RPS31 changes structure in vitro- we also performed RNA structure probing in vivo to determine structural changes in the presence of FMN. However in vivo targeted RNA structure probing is technically very difficult because RPS31 transcript levels is too low to be detected on a radioactive gel, without amplification. As such, we only managed to obtain one replicate after many attempts. We have removed this data from the manuscript.

REVIEWER 4: It is appropriate that these data have been removed.

2.) Fig 9 is not an appropriate equation for the data. The authors need a curve that doesn't go through the origin but allows a y-intercept of any value, e.g. ~ 1 . Here the 1/2 value will be >100 μM FMN, not 50 μM .

We thank the reviewer for pointing this out and agree that we are ignoring the background signal in the extrapolated curve. Our recalculations show a K_d of $\sim 137 \mu\text{M}$ for RPS31-FMN interactions and we have updated this value in the manuscript (Extended data 11c).

REVIEWER 4: This is a step in the right direction and the value looks to be appropriate from graphical analysis of Extended Data 11c. However, the authors need to show data and a smooth fit. What they currently show in Extended Data 11c seems to be a connect-the-dots plot of the data, given it is non-monotonic, with no fit. They say “Our recalculations show a K_d of $\sim 137 \mu\text{M}$...” What do they mean by “recalculation”? Is there any calculation here or just a best guess at where $\frac{1}{2}$ max is? Please replace this with data as points and a smooth fit to a proper K_d theoretical curve using a non-linear fitting program.

We thank the reviewer for his comments. We modelled our data using non-linear regression in Prism, using the analysis for single site binding saturation. The K_d is now estimated to be $187 \mu\text{M}$. We have updated this figure as Supplementary Figure 11d in the manuscript.

Figure 1. Plot showing the relative intensity of band by in-line probing under increasing concentrations of FMN. A best fit curve is plotted based on non-linear regression to identify the K_D of binding. (Supplementary Figure 11d)

$$\text{Equation: } y = 0.1404x / (187.1 + x)$$

4.) This was the main concern and is not addressed. The biology just doesn't calculate.

We thank the reviewer for his comments. We have three examples of gene regulation by FMN (one at the transcriptional level for *B. subtilis* tmRNA, and two at the protein level for *C. albicans* ATP1 and RPS31).

REVIEWER 4: What do the authors mean when they write above that “We have three examples” ? Who are the “We” in this sentence? The field or the authors lab? I believe they mean the field and if so the writing here is potentially misleading.

By “we”, we mean us as authors and not that of the field. The three examples are *B. subtilis* tmRNA, *C. albicans* ATP1 and RPS31.

While identifying riboswitches is one of the aims for designing PARCEL, we note that PARCEL is a platform to find naturally occurring RNA aptamers in virtually any transcriptome, towards any metabolite.

REVIEWER 4: The implication in the writing here via “naturally occurring” is that the aptamer is an in vivo aptamer, but the data as is do not support this. To be publishable, this statement should be revised to not imply that these are naturally occurring without direct demonstration in vivo as such.

We respectfully disagree with the reviewer. As we identified these RNA aptamers from naturally occurring transcriptomes in organisms, we mean that the RNA aptamers are found in existing transcriptomes, in contrast to synthetic aptamers which are man-made/ designed and are not naturally found in any organisms.

Some of the new aptamers may be riboswitches with important gene regulatory functions, while others can be used in metabolite engineering to create higher affinity RNA aptamers for intracellular molecular sensing. As such, we believe that detailed mechanistic studies are beyond the scope of this paper as they will require significant and extended investigations to understand the role of RNA sensors in global cell biology.